# Dietary *Isatidis Root* Residue Improves Diarrhea and Intestinal Function in Weaned Piglets

**DOI:** 10.3390/ani14192776

**Published:** 2024-09-26

**Authors:** Zhong Chen, Zenghao Yan, Siting Xia, Kaijun Wang, Qi Han, Miao Zhou, Deqin Wang, Jie Yin, Yulong Yin

**Affiliations:** 1Animal Nutritional Genome and Germplasm Innovation Research Center, College of Animal Science and Technology, Hunan Agricultural University, Changsha 410128, China; cz010808@163.com (Z.C.); siting_hsia@stu.hunau.edu.cn (S.X.); kj-wang@foxmail.com (K.W.); 18404985053@163.com (Q.H.); zhoumiao@stu.hunau.edu.cn (M.Z.); yinjie@hunau.edu.cn (J.Y.); 2Hutchison Whampoa Guangzhou Baiyunshan Chinese Medicine Co., Ltd., Guangzhou 510515, China; yanzengh@yeah.net (Z.Y.); wdq2008@163.com (D.W.); 3Institute of Subtropical Agriculture, Chinese Academy of Sciences, Changsha 410125, China

**Keywords:** *Isatidis Root* residue, weaning stress, diarrhea, intestinal morphology, tight junction, gut microbiota

## Abstract

**Simple Summary:**

Traditional Chinese herbal resources are abundantly available in China and have garnered significant attention owing to their distinctive pharmacological attributes, such as their antibacterial, anti-inflammatory, and antiviral properties, and their ability to regulate their host’s health. Nevertheless, a considerable amount of residue remains after the extraction of these herbal medicines. Presently, limited research on the utilization of this residue has led to both resource wastage and environmental pollution. Numerous studies have indicated that the extracted residue of the *Isatidis Root* still contains various organic components, including amino acids, alkaloids, flavonoids, and other nutrients. Consequently, this study’s objective was to explore the potential of *Isatidis Root* residue as an unconventional feed resource. Our findings revealed that *Isatis Root* residue positively impacted the diarrhea rate and intestinal health of weaned piglets.

**Abstract:**

Weaning stress can trigger diarrhea, cause intestinal damage, and disrupt the intestinal flora of piglets, ultimately resulting in retarded growth or even the death of the animals. Traditional Chinese medicine residues encompass numerous bioactive compounds and essential nutrients; however, their efficient utilization remains a challenge. Consequently, our study sought to explore the impact of traditional Chinese medicine residues, specifically *Isatidis Root* residue (IRR), on the growth performance, intestinal function, and occurrence of weaning diarrhea in newly weaned piglets. Forty healthy, castrated Duroc × Landrace × Yorkshire males, weaned at 21 days old and exhibiting similar body conditions, were randomly allocated into five groups, with eight piglets in each group. The results indicated that the dietary inclusion of IRR at concentrations ranging from 0.5% to 4.0% notably decreased the incidence of diarrhea in weaned piglets compared to the control group (*p* < 0.05). Serum LDL-C and globulin (GLB) contents were reduced in response to dietary IRR concentrations (0.5% to 4.0%), while serum albumin (ALB) and albumin/globulin (A/G) contents were enhanced (*p* < 0.05). Dietary 0.5%, 1.0%, and 2.0% IRR resulted in significant increases in villus height (VH) and villus height/crypt depth (V/C) ratios in the jejunum, V/C ratios in the ileum, and the number of villi goblet cells both in the jejunum and ileum. IRR also led to a significant decrease in the crypt depth (CD) of the jejunum and ileum (*p* < 0.05). Furthermore, the expression of IL-6 in the jejunum was significantly increased in IRR-fed piglets (0.5% to 4.0%) (*p* < 0.05). IRR demonstrated inhibitory effects on harmful bacteria in the gastrointestinal microbiome, including *Campylobacter*, *Actinobacillus minor*, and *Ralstonia pickettii*, indicating its broad-spectrum bacteriostatic properties. In conclusion, dietary IRR alleviated diarrhea in weaned piglets and improved gut function and microbial compositions.

## 1. Introduction

During the weaning process, piglets encounter physiological and environmental challenges that disrupt the balance of their gut microbiota and immune system, ultimately leading to weaning stress [1]. The occurrence of diarrhea in weaned piglets is a concern, with rates reaching up to 20% [2]. Despite attempts to manage this using a combination of antibiotics and other drugs, their careless or haphazard administration has raised concerns about antibiotic resistance, drug residues, and environmental contamination [3]. Therefore, it is crucial to explore new alternatives to antibiotics for managing weaning stress and diarrhea in piglets.

*Isatis indigotica* Fort. (commonly known as woad) belongs to the cruciferous family [4]. Its dried roots, which have a rich history spanning thousands of years in traditional Chinese medicine, are renowned for their heat-clearing, detoxifying, blood-cooling, and throat-soothing properties [5]. Contemporary research has revealed that the roots of *Isatis indigotica* exhibit antiviral and anti-inflammatory effects. In southern China, this herbal medicine is widely consumed as a tea and frequently prescribed by clinicians for treating colds, fevers, and sore throats [6]. Recently, several *Isatis Root* extract products, such as granules, capsules, oral liquids, and herbal tea drinks, have emerged on the market. Consequently, significant amounts of Isatis residue are generated annually in China [7].

In an industrial context, residues from traditional Chinese medicinal herbs are often regarded as waste [8]. This practice not only squanders valuable resources but also poses environmental threats [4]. Due to limitations in extraction techniques, these residues still contain numerous nutrients and bioactive compounds [9]. Previous studies have identified multiple active ingredients in *Isatis Root*, notably including chlorogenic acid, rutin, and quercetin, alongside various amino acids and alkaloids [10]. Interestingly, chlorogenic acid, rutin, and quercetin are also prominent active components in mulberry leaves, whereas wild chrysanthemum is characterized by chlorogenic acid, luteolin, and monoglycoside [11,12]. These bioactive substances are known to exert diverse biological effects, such as promoting animal growth, alleviating diarrhea, exhibiting antioxidant properties, reducing stress, enhancing gut microbiota, and modulating immune responses [13,14]. Hence, utilizing medicinal herb residues in animal production represents a commendable practice and constitutes a crucial aspect of recycling these valuable resources.

Although previous research has indicated the potential of the *Isatidis Root* in mitigating weaning stress among piglets, there remains a paucity of data regarding the impact of *Isatidis Root* residue (IRR) on weaning stress [15,16]. Consequently, the aim of this study was to assess the influence of varying concentrations of IRR on the health status and gut microbiota of weaned piglets. By doing so, we aspire to introduce a novel approach for easing weaning stress in piglets, advance the utilization of IRR as animal feed, and ultimately mitigate environmental stressors.

## 2. Materials and Methods

This animal study was reviewed and approved by the Hunan Agricultural University Institutional Animal Care and Use Committee (202105). Written informed consent was obtained from the owners for the participation of their animals in this study.

### 2.1. Materials

The IRR, with a moisture content of 6.64 ± 0.34%, was supplied by Guangzhou Baiyunshan Xingqun Pharmaceutical Co., Ltd. (Guangzhou, China). The preparation method for the residue is described below: Take an *Isatis Root* with a mass ratio of 500:175:80. Then, add water at a ratio of 10 times the weight of the sample. Decoct the mixture at 100 °C for 1.5 h, with a total of two decoctions. Next, filter and dry the mixture to obtain the IRR. Finally, incorporate the residue into the feed according to the specified formula ratio.

### 2.2. Quantitative Analysis of Active Ingredients in Residues

Through careful optimization of the chromatographic and mass spectrometric parameters, the definitive LC-MS detection conditions were established as the following: an Agilent Eclipse XDB-C18 column (dimensions: 2.1 mm × 150 mm, 1.8 μm particle size, Agilent Technology Co., Ltd., Beijing, China) with a mobile phase consisting of 0.1% formic acid in water (A) and acetonitrile (B); a gradient elution program of 2–30% B for 0–8 min, 30–95% B for 8–25 min, maintained at 95% B for 25–30 min, and returned to 2% B for 30.1–35 min; a column temperature maintained at 35 °C; an autosampler temperature at room temperature (25 ± 2 °C); a flow rate set to 0.3 mL/min; an injection volume of 1 μL; and data acquisition using a Diode Array Detector (DAD) with full-wavelength scanning from 210 to 400 nm. The high-resolution mass spectrometry (HRMS) data acquisition conditions included an Agilent Dual AJS ESI source, positive ion full scan with a range of *m*/*z* 100 to 1000, sheath gas temperature set at 350 °C, sheath gas flow rate of 11.0 L/min, drying gas temperature of 345 °C, drying gas pressure of 45 psi, drying gas flow rate of 10 L/min, capillary voltage of 4000 V, fragmentor voltage of 135 V, and the use of Agilent MassHunter Acquisition Workstation Software (version B.08.00). The high-resolution tandem mass spectrometry (HRMS/MS) data acquisition conditions involved data collection utilizing the target MS/MS mode, ion source parameters mirroring those of the MS mass spectrometry, and collision energy selected within the 5 to 45 eV range, depending on the molecular weight and structural stability of the target compound (Table 1 and Table 2).

### 2.3. Animals and Dietary Treatments

Forty 21-day-old Duroc × Landrace × Yorkshire weanling piglets (sourced from Changde Hanshou Tianxin Animal Husbandry Co., Ltd. (Changde, China.) with comparable bodily conditions were randomly assigned to five groups, each comprising eight replicates with one pig per replicate. The environmental conditions within the pig house were meticulously maintained, ensuring a relative humidity of 50% to 60% and a stable temperature range of 23 to 25 °C. Each pig was housed individually, and feeding took place at four scheduled times daily: 6 a.m., 10 a.m., 2 p.m., and 6 p.m. Piglets in the control group received a standard basal diet. Simultaneously, each treatment group was provided with the same basal diet, but varying proportions of IRR were incorporated as the following: 0.5%, 1.0%, 2.0%, and 4.0%, respectively. Prior to use, the IRR was pulverized using a 60-mesh sieve. The composition of the basal diet (Table 3) adhered closely to the NRC 2012 guidelines, ensuring that the nutrient content conformed to the NRC’s recommendations.

### 2.4. Sample Collection

At the conclusion of the experiment, blood samples were collected from the anterior vena cava of all piglets using a standard 10 mL blood collection tube. The samples were allowed to rest at room temperature for 20 min before being centrifuged at 845 rcf (g) for 10 min. The serum layer was then carefully collected into sterile, frozen tubes and stored in liquid nitrogen at −80 °C. The piglets were anesthetized with 3% sodium pentobarbital at a dosage of 25 mg/kg and euthanized via exenteration. The abdominal cavity was then opened to separate the viscera and intestine. Two 2 cm segments of intestinal tissue were excised, one from the anterior segment of the jejunum and one from the posterior segment of the ileum. One segment from each location was placed in a 4% paraformaldehyde solution for histological analysis, while the other was promptly rinsed with normal saline, placed in a 2 mL sterile cryostat, and stored in liquid nitrogen. Additionally, a 10 cm section of the anterior colon was removed, and a small incision was made with a scalpel. The ends were then tied together with a string, and the gut contents were transferred into a 2 mL sterile cryostat and stored in liquid nitrogen. Lastly, the ileal digesta was also collected into a 2 mL sterile cryostat and stored in liquid nitrogen.

### 2.5. Growth Performance

On day 1 and day 21, each piglet was weighed, and the daily amount of food supplied and remaining was recorded to calculate the average daily feed intake (ADFI), average daily gain (ADG), and feed-to-weight ratio (F/G). To assess diarrhea, the piglets were scored daily at approximately 15:00 using the following criteria: 1 indicated solid, hard stool; 2 represented slightly loose stool; 3 denoted soft, partially formed stool; 4 corresponded to semiliquid stool; and 5 signified stool–water separation, watery, and unformed stool.

### 2.6. Organ Index

After slaughter, the piglets’ organs were separated, and the surface liquid of each organ was promptly blotted dry using filter paper before being weighed. The organ index was calculated using the following formula: organ index = organ weight (g)/live weight of piglets (kg).

### 2.7. Serum Biochemical Parameter Analysis

The concentrations of serum total protein, albumin, total bile acid, glucose, triglyceride, urea nitrogen, total cholesterol, high-density lipoprotein cholesterol, low-density lipoprotein cholesterol, alanine aminotransferase, aspartate aminotransferase, alkaline phosphatase, and lactate dehydrogenase were determined using an automatic biochemical analyzer (KHB 450, Shanghai Kehua Bioengineering Co., Ltd., Shanghai, China) and its corresponding reagents, following the method described by Shi et al. [17].

### 2.8. Immunoglobulin Analysis

The levels of serum IgA, IgG, and IgM were determined using ELISA kits provided by Jiangsu Meimian Industrial Co., Ltd. (Yancheng, China), while the content of sIgA in ileal digesta was measured using ELISA kits obtained from Quanzhou Ruixin Biological Technology Co., Ltd. (Quanzhou, China). All detection procedures were performed in accordance with the kit instructions, as described by Yu et al. [18].

### 2.9. Intestinal Morphology

Following the methodology outlined in our previous study [19], intestinal tissue was fixed in a 4% paraformaldehyde solution, subsequently cut and dehydrated, embedded in paraffin, sectioned, dewaxed, and stained with hematoxylin and eosin. After further dehydration, the tissue sections were sealed with neutral glue. Images were captured using a microscope imaging system (Carl Zeiss, Oberkochen, Germany) to measure intestinal villus height (VH) and crypt depth (CD). The ratio of villus height to crypt depth (V/C) was calculated for five randomly selected fields per tissue slice.

### 2.10. AB-PAS Staining

Subsequent to our previous study, tissue paraffin blocks were sliced, deparaffinized, and rehydrated. They were then stained in accordance with the instructions provided by an AB-PAS test kit (Nanjing Jiancheng Bioengineering Institute, Nanjing, China), followed by sealing with neutral resin. The stained tissue slices were ultimately observed and photographed using a Carl Zeiss Microimaging System. For goblet cell number determination, ten intact villi and crypts were selected from each piglet sample, and the results are expressed as the number of goblet cells per villus.

### 2.11. Quantitative Real-Time PCR Analysis

Intestinal tissue was frozen and ground in liquid nitrogen. Subsequently, total RNA was isolated using RNAiso Plus (TaKaRa, Dalian, China), followed by reverse transcription using a PrimeScript™ RT Reagent Kit with gDNA Eraser (TaKaRa, Dalian, China). Fluorescence quantification was then carried out using TB Green^®^ Fast qPCR Mix in a LightCycler 480 System II (Roche, Shanghai, China). The primers utilized in this study were designed based on porcine sequences (Table 4). PCR cycles and relative expression assays were conducted in accordance with our previous study [20].

### 2.12. Microbial Analysis

Total microbial genomic DNA was extracted from the colonic digesta using a Power Fecal DNA Extraction Kit (MOBIO, Carlsbad, CA, USA). Universal primers 341F (5′-ACTCCTACGGGAGGCAGCAG-3′) and 806R (5′-GGACTACHVGGGTWTCTAAT-3′) were utilized to amplify the V3–V4 region of the 16S rRNA gene. Following purification, the PCR product was employed to construct a library using the NEB NextB UltraTM DNA Library Prep Kit from New England Biolabs, Inc. (Ipswich, MA, USA), specifically the Lumina Library Construction Kit. The constructed library was quantified and analyzed using a Qubit. Once deemed suitable, Illumina MiSeq PE300 (Illumina, Sandiego, CA, USA) was utilized for on-machine sequencing. The raw sequences were analyzed using QIIME, version 1.7.0. Initial reads underwent mass filtering, denoising, and assembly, with chimera sequences removed according to the method described by Deblur et al. [21]. Only ASVs with at least 2 reads and present in more than 2 samples were retained. PICRUSt2 was employed to predict gut microbiota function. All data were analyzed using the NovoMagic cloud platform https://magic.novogene.com (accessed on 11 April 2023).

### 2.13. Statistical Analysis

Data on the growth performance, organ index, serum parameters, and RT−qPCR were analyzed using one-way ANOVA with SPSS 26.0 statistical software. Multiple comparisons were conducted using Duncan’s method, with *p* < 0.05 considered as indicating a significant difference and *p* < 0.01 considered as indicating an extremely significant difference. The results are presented as the mean ± SD.

## 3. Results

### 3.1. Growth Performance and Diarrhea Score

The results indicated that, in comparison to the control group, piglets in all IRR groups exhibited no significant alterations in final weight, weight gain, feed intake, or feed coefficient (*p* > 0.05). Notably, both ADFI and diarrhea rates were significantly reduced compared to the control group (*p* < 0.05) (Figure 1A).

Regarding the organ index, a notable elevation in the spleen organ index was exclusively detected in piglets administered the 1.0% IRR diet relative to the control piglets (*p* < 0.05). However, this increase was not evident in the other treatment groups when compared to the control (Figure 1B).

### 3.2. Serum Biochemical Parameters

The results revealed that serum TC and TG concentrations were significantly elevated in both the IRR 1.0% and 4.0% groups. Furthermore, the ALB level and albumin/globulin (A/G) ratio were notably increased across all IRR treatment groups (*p* < 0.05). Conversely, GLB and LDL-C levels were significantly reduced in all IR-treated groups (*p* < 0.05). Additionally, the incorporation of 2.0% IRR into the diet led to a marked decrease in total serum bile acids compared to the control group. Notably, all IRR dose groups exhibited significant reductions in LDL-C and globulin (*p* < 0.05), and IgA was significantly decreased in the 1% IRR dose group. However, there were no statistically significant differences in glucose (GLU), HDL-C, AST, ALP, LDH, TP, IgG, IgM, or BUN levels among the groups (*p* > 0.05) (Figure 2). 

### 3.3. Intestinal Morphology

Jejunal and ileal samples were utilized to assess intestinal morphology via HE staining. The results demonstrated that dietary supplementation with 0.5% and 1.0% IRR significantly enhanced VH and V/C and decreased CD in both the jejunum and ileum (*p* < 0.05) (Figure 3). AB-PAS staining was employed for the evaluation of intestinal goblet cells (Figure 4). The findings revealed a notable increase in goblet cell numbers in both the jejunum and ileum of piglets when fed diets containing 0.5% and 1% IRR (*p* < 0.05).

### 3.4. Expressions of Tight Barrier and Inflammation-Related Genes

The impact of IRR on intestinal health at the molecular level was assessed by examining the expression of tight barrier and inflammation-related genes. The expression levels of occludin, ZO-1, IL-10, IL-1β, and TNF-α remained unaffected in the jejunum and ileum of piglets fed with IRR concentrations ranging from 0.5% to 4.0%. In the jejunum, dietary supplementation with 1.0% IRR showed a slight tendency to elevate Claudin1 expression (*p* > 0.05). Piglets fed with 1.0%, 2.0%, and 4.0% IRR demonstrated reduced expression of IL-6 in the jejunum compared to the control group (*p* < 0.05) (Figure 5A). In the ileum, dietary supplementation with 0.5% IRR displayed a trend towards increased Claudin1 expression (*p* > 0.05) (Figure 5B).

### 3.5. Intestinal Bacterial Compositions

The α-diversity was evaluated in this study through the analysis of the Chao1, ACE, Shannon, and Simpson indices (Figure 6A). However, no significant changes were observed in the α-diversity of the IRR-treated piglets (*p* > 0.05).

Further analysis was conducted on the gut bacterial compositions at the phylum, genus, and species levels (Figure 6B–D). At the phylum level, the primary colonic microbial phyla were identified as Firmicutes, Bacteroidetes, and Proteobacteria. Nonetheless, the relative abundances of Firmicutes, Bacteroidetes, and Proteobacteria did not exhibit significant changes in response to dietary IRR concentrations ranging from 0.5% to 4.0% (*p* > 0.05). However, dietary IRR (0.5–4.0%) resulted in a substantial decrease in the abundances of Campylobacterota, Actinobacteriota, Verrucomicrobiota, and Deferribacteres, while a 4.0% IRR concentration increased the abundances of Spirochaetota and Fusobacteriota (*p* < 0.05) (Figure 6B).

At the genus level, dietary supplementation with 1.0%, 2.0%, and 4.0% IRR caused a significant decrease in the abundance of *Campylobacter*, while 0.5%, 2.0%, and 4.0% IRR significantly decreased the abundances of *Blautia*, *Agathobacter*, and *Subdoligranulum* (*p* < 0.05). A 4.0% IRR concentration resulted in a marked reduction in *Lactobacillus* and *Clostridium sensu stricto*, while it increased the abundance of *Treponema* compared to the control group (*p* < 0.05) (Figure 6C).

At the species level, dietary supplementation with IRR (0.5–4.0%) significantly decreased the abundances of *Eubacterium coprostanoligenes*, *Lactobacillus reuteri*, *Actinobacillus minor*, *Ralstonia pickettii*, and *Lactobacillus murinus*. A 4.0% IRR concentration significantly decreased the abundances of *Lactobacillus amylovorus* and *Lactobacillus salivarius*. In addition, 0.5%, 2.0%, and 4.0% IRR significantly reduced the abundance of Olsenella_sp_GAM18 (*p* < 0.05) (Figure 6D).

## 4. Discussion

Traditional Chinese herbs (TCHs) have been extensively utilized as feed supplements to promote healthy aquacultural practices within the animal industry [22]. For example, research has demonstrated that dietary supplementation comprising *Paeonia lactiflora*, licorice, dandelion, and tea polyphenols notably augments the average daily gain in weaned pigs [23]. Similarly, weaned pigs treated with fermented IRR have exhibited both a greater average daily gain and improved feed efficiency [24]. However, the present study did not observe any improvement in growth performance following IRR treatment in the pig model. This discrepancy may be attributed to the high lignin and fiber content of IRR, suggesting that dietary supplementation with IRR does not necessarily negatively impact the growth performance of weaned piglets. This finding is consistent with the observations made by Liu et al. in their study on pigs [25].

During the post-weaning period, a high incidence of diarrhea is prevalent among pigs [26]. In our study, dietary supplementation with IRR reduced the frequency of diarrhea in pigs compared to the control group. Several previous studies have also reported the positive effects of dietary Chinese herbal supplementation on the diarrhea frequency of weaned pigs. For instance, Huang et al. [27] observed this phenomenon and attributed it to suppressed *Escherichia coli* growth in the gut. Furthermore, Zhao et al. [28] reported significantly increased immune function when weaned pigs were fed diets containing 100 mg/kg *Forsythia suspensa* extract. The decreased relative abundance of harmful bacteria in the intestine and increased gut villus structure, intestinal crypt, and goblet cell count in weaned pigs consuming IRR in our study may explain the positive effects of IRR supplementation on diarrhea frequency and nutrient digestibility. These factors may collectively regulate the gut microbial community and benefit the gut health of the host.

Blood biochemical indices can serve as indicators of growth performance and metabolism in animals [29,30]. The results of this study indicated that dietary IRR significantly increased serum total cholesterol (TC) and triglyceride (TG) levels and decreased serum LDL-C levels in weaned piglets. However, previous experimental studies have demonstrated that the dietary supplementation of the *Raphani seed*, *Atractylodes rhizoma*, *Coicis seed*, *poria*, hawthorn, and *Codonopsis rhizoma* can significantly lower the levels of TC, TG, and LDL-C in piglets [31]. This discrepancy may be attributed to the multiple amino acids and other compounds contained in IRR. For instance, Aoyama et al. [32] found that feeding mice excessive amounts of amino acids led to fat accumulation. Serum TC and TG concentrations reflect abnormal lipid metabolism and fat accumulation, while serum LDL-C concentrations are strongly associated with the risk of atherosclerotic cardiovascular disease [33,34]. Additionally, we found that dietary IRR significantly increased serum albumin (ALB) levels and decreased serum globulin (GLB) levels in weaned piglets. Previous experiments have shown that the addition of *Folium mori* also increased ALB levels and decreased GLB levels in piglets [25]. Serum ALB concentration reflects the antioxidant capacity and protein synthesis ability of the body, while during long-term inflammation, serum GLB concentration increases and, to a certain extent, reflects liver injury [35,36,37]. Therefore, IRR may be beneficial to liver metabolism and, consequently, the health of weaned piglets.

The structure of gut villi and the integrity of the intestinal barrier play a crucial role in intestinal function, encompassing nutrient digestibility, absorption, and protection against pathogenic infection [38]. Zhao et al. [39] reported that dietary dandelion root extract administered for 28 days enhanced the apparent total tract digestibility of dry matter, nitrogen, and gross energy in pigs, indicating improved gut absorptive function. Furthermore, previous research has demonstrated that a mixture of herbal extracts (golden-and-silver honeysuckle, huangqi, duzhong leaves, and dangshen) administered for 14 days augmented gut villus structure and barrier integrity [40]. Similarly, our results suggest that IRR increased gut villus structure, intestinal crypt depth, and goblet cell counts. However, there was no significant change in tight junction proteins. Notably, we found that dietary IRR enhanced Claudin-1 expression in the jejunum but decreased Claudin-1 expression in the ileum. Therefore, further studies are necessary to elucidate the role of IRR in different intestinal segments of piglets.

TCHs, such as licorice, *Codonopsis pilosula*, *Astragalus membranaceus*, *Atractylodes*, *Angelica sinensis*, *Rhizoma cimicifugae*, *Radix bupleuri*, dried tangerine peel, ginger, and jujube, have been extensively reported to enhance the gut microbiome [41,42]. However, in the present study, we did not observe any improvement in α- or β-diversity following IRR treatment in a pig model. Concurrently, dietary IRR led to a reduction in the levels of *Lactobacillus*, *Agathobacter*, *Blautia*, *Subdoligranulum*, and *Campylobacter*. Prior research has indicated that traditional Chinese herbal administration decreases harmful bacteria and increases beneficial bacteria, which contrasts with the findings of the current study, where beneficial bacteria were reduced at the genus level [23]. Zhao et al. [39] also confirmed that the relative abundance of beneficial bacteria in piglets’ gut microbiota was not altered after CHM administration. This discrepancy might be attributed to the composition and physicochemical properties of the IRR used in this study. Gut microbiota can be rapidly influenced by dietary nutrients, which further mediate the host metabolism and physiological response [43]. At the species level, *Actinobacillus minor* and *Ralstonia pickettii* are commonly recognized as Gram-negative bacterial pathogens that can cause respiratory tract infection and sepsis in livestock and poultry [44,45]. Our findings revealed that dietary IRR supplementation significantly decreased the relative abundances of *Actinobacillus minor* and *Ralstonia pickettii*. Additionally, *Eubacterium coprostanoligenes* was also significantly reduced. Previous studies have shown that *Eubacterium coprostanoligene* plays a crucial role in maintaining the intestinal barrier and produces metabolites such as short-chain fatty acids (SCFAs) through its metabolic activity [46]. Interestingly, *Eubacterium coprostanoligene* has the ability to participate in metabolizing cholesterol, converting easily absorbed cholesterol into poorly absorbed cholesterol [47]. We speculate that this may explain the observed increase in the serum biochemical measures of total cholesterol (TC) and triglycerides (TG) and the decrease in LDL-C in this trial.

## 5. Conclusions

In conclusion, dietary supplementation with IRR has demonstrated effectiveness in mitigating early weaning-associated diarrhea, bolstering intestinal barrier integrity and immune response, as well as modulating intestinal microbiota composition. The most pronounced effects were observed with 0.5–1% supplementation. These results suggest the promise of IRR as an innovative feed component or functional additive. However, it should be emphasized that this investigation constitutes a preliminary study, and IRR exhibited no notable impact on the piglet weight gain or feed conversion ratio. Future studies ought to concentrate on conducting extensive, long-term feeding experiments to further elucidate the function of IRR.

## Figures and Tables

**Figure 1 animals-14-02776-f001:**
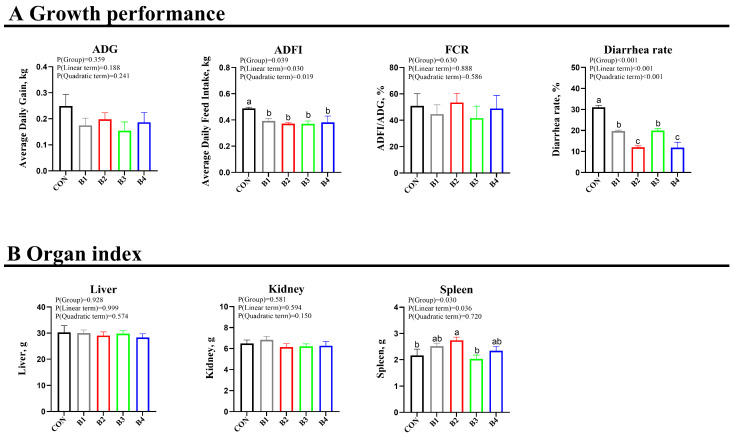
Effect of IRR on the growth performance of weaning piglets. (**A**) Growth performance, (**B**) Organ index. ADG, average daily gain; ADFI, mean daily feed intake; FCR, ratio of average daily gain intake to average daily feed. CON = basal diet, B1 = basal diet supplemented with 0.5% IRR, B2 = basal diet supplemented with 1.0% IRR*e*, B3 = basal diet supplemented with 2.0% IRR, and B4 = basal diet supplemented with 4.0% IRR. In the regions marked as a, b, and c within the figure, no significant differences were observed among groups denoted by the same letter. Conversely, statistically significant differences were evident between groups designated by different letters.

**Figure 2 animals-14-02776-f002:**
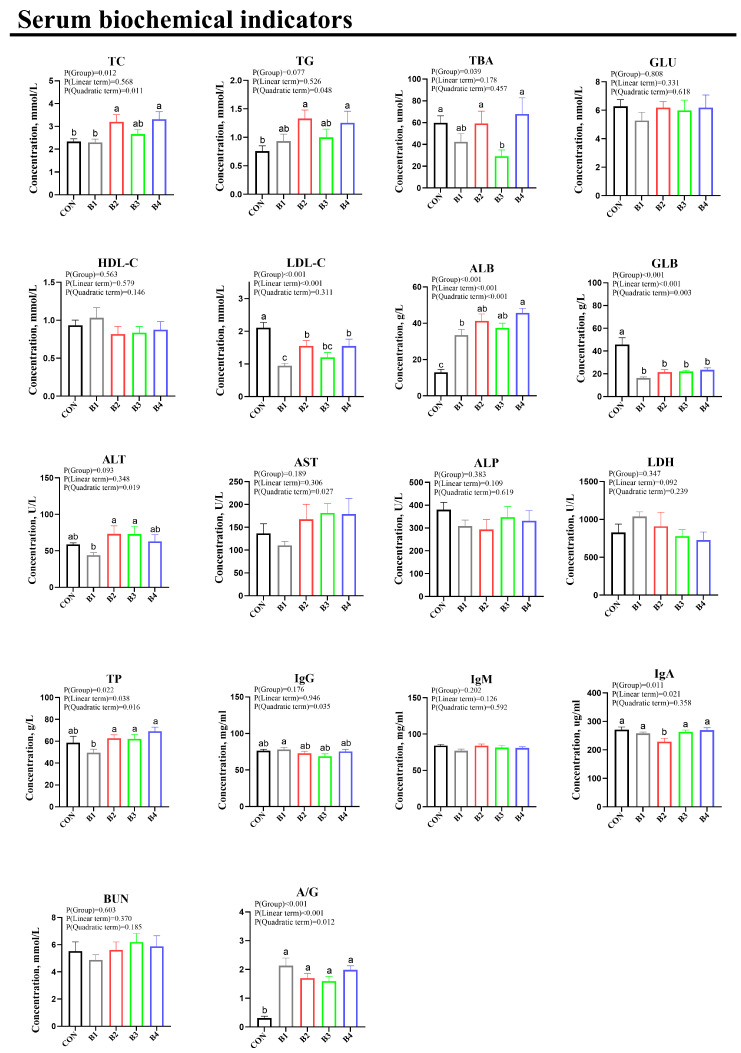
Effect of IRR on serum biochemical parameters of weaning piglets. Serum biochemical indicators. TC = total cholesterol; GLU = glucose; TBA = total bile acid; HDL-C = high-density lipoprotein cholesterol; LDL-C = low-density lipoprotein cholesterol; ALT = alanine aminotransferase; AST = aspartate aminotransferase; ALP = alkaline phosphatase; LDH = lactate dehydrogenase; TP = total protein; ALB = albumin; GLB = globulin; A/G = albumin to globulin ratio; BUN = blood urea nitrogen; IgG = immunoglobulin G; IgM = immunoglobulin M; IgA = immunoglobulin A. CON = basal diet, B1 = basal diet supplemented with 0.5% *Isatidis Root Residue*, B2 = basal diet supplemented with 1.0% *Isatidis Root Residue*, B3 = basal diet supplemented with 2.0% *Isatidis Root Residue*, and B4 = basal diet supplemented with 4.0% *Isatidis Root Residue*. In the regions marked as a, b, and c within the figure, no significant differences were observed among groups denoted by the same letter. Conversely, statistically significant differences were evident between groups designated by different letters.

**Figure 3 animals-14-02776-f003:**
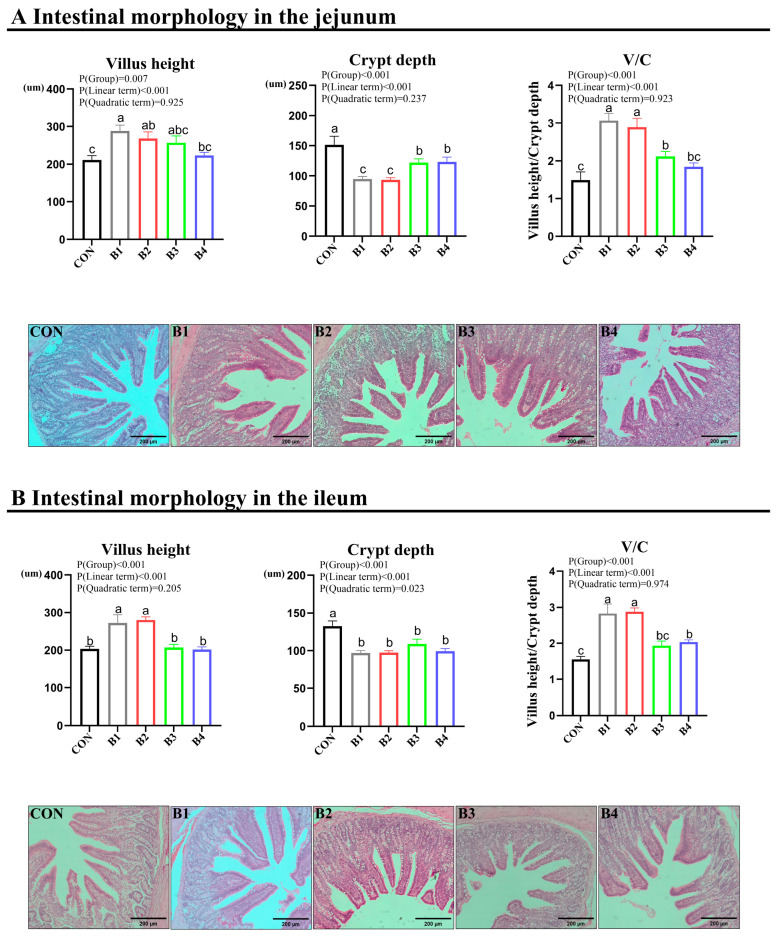
Effect of IRR on immunity and intestinal morphology. (**A**) Intestinal morphology in the jejunum, (**B**) Intestinal morphology in the ileum. V/C = villus height/crypt depth. B1 = basal diet supplemented with 0.5% IRR, B2 = basal diet supplemented with 1.0% IRR, B3 = basal diet supplemented with 2.0% IRR, and B4 = basal diet supplemented with 4.0% IRR. In the regions marked as a, b, and c within the figure, no significant differences were observed among groups denoted by the same letter. Conversely, statistically significant differences were evident between groups designated by different letters.

**Figure 4 animals-14-02776-f004:**
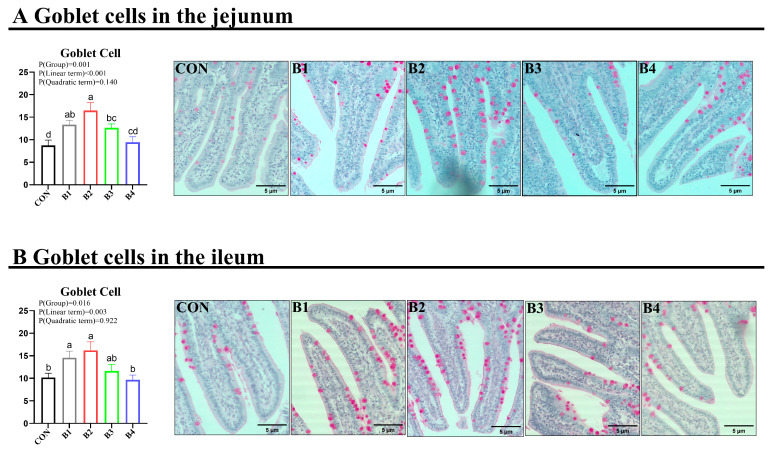
Effect of IRR on goblet cells in the jejunum and ileum. (**A**) Goblet cells in the jejunum, (**B**) Goblet cells in the ileum. B1 = basal diet supplemented with 0.5% IRR, B2 = basal diet supplemented with 1.0% IRR, B3 = basal diet supplemented with 2.0% IRR, and B4 = basal diet supplemented with 4.0% IRR. In the regions marked as a, b, c and d within the figure, no significant differences were observed among groups denoted by the same letter. Conversely, statistically significant differences were evident between groups designated by different letters.

**Figure 5 animals-14-02776-f005:**
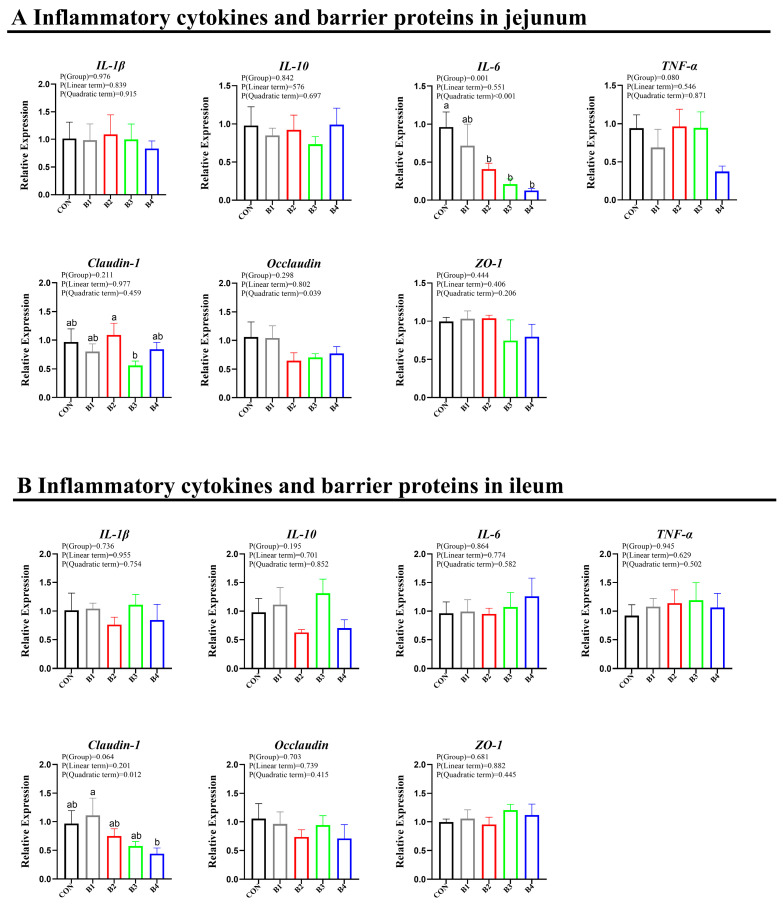
Effect of IRR on intestinal inflammation and the intestinal barrier. (**A**) Inflammatory cytokines and barrier proteins in the jejunum, (**B**) Inflammatory cytokines and barrier proteins in the ileum. IL-β = interleukin-β, IL-10 = interleukin-10, IL-6 = interleukin-6, TNF-α = tumor necrosis factor-α. B1 = basal diet supplemented with 0.5% IRR, B2 = basal diet supplemented with 1.0% IRR, B3 = basal diet supplemented with 2.0% IRR, and B4 = basal diet supplemented with 4.0% IRR. In the regions marked as a and b within the figure, no significant differences were observed among groups denoted by the same letter. Conversely, statistically significant differences were evident between groups designated by different letters.

**Figure 6 animals-14-02776-f006:**
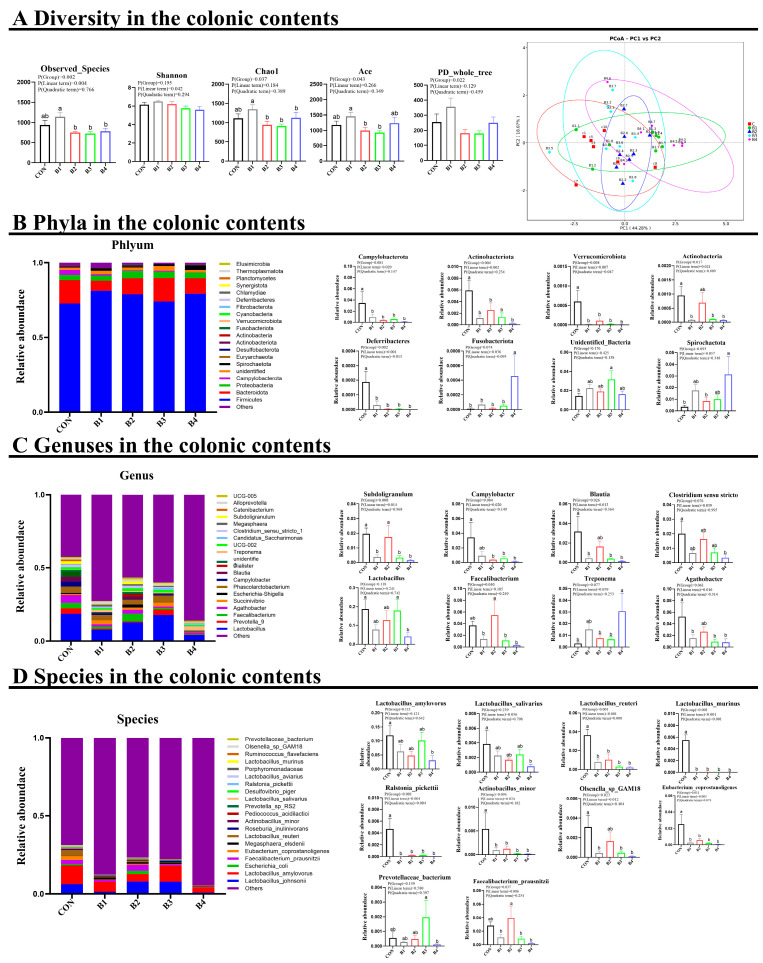
Effect of IRR on the gut microbiota of early weaning piglets. (**A**) Diversity in the colonic contents, (**B**) Phyla in the colonic contents, (**C**) Genera in the colonic contents, (**D**) Species in the colonic contents. B1 = basal diet supplemented with 0.5% IRR, B2 = basal diet supplemented with 1.0% IRR, B3 = basal diet supplemented with 2.0% IRR, and B4 = basal diet supplemented with 4.0% IRR. In the regions marked as a and b within the figure, no significant differences were observed among groups denoted by the same letter. Conversely, statistically significant differences were evident between groups designated by different letters.

**Table 1 animals-14-02776-t001:** Names and contents of compounds identified in the IRR.

No.	L-Arg	Guanine	L-Phe	Epigoitrin	Deoxyvasicinone	3-Indolylacetonitrile	Indigo	Indirubin
1	1.26	1.84	0.34	0.45	0.23	0.67	2.88	0.33
2	0.98	0.98	0.22	0.49	0.26	0.70	1.58	0.29
3	0.87	0.74	0.20	0.48	0.24	0.77	1.56	0.28
4	1.01	0.84	0.20	0.43	0.22	0.63	1.56	0.25
5	1.46	1.27	0.22	0.37	0.19	0.54	2.28	0.22
6	1.43	1.55	0.23	0.36	0.17	0.49	2.84	0.21
7	1.03	0.64	0.12	0.31	0.12	0.43	1.43	0.14
8	1.43	0.64	0.16	0.41	0.17	0.40	1.55	0.18
9	1.39	0.78	0.13	0.33	0.12	0.39	1.55	0.15
10	1.28	0.65	0.11	0.35	0.11	0.40	1.69	0.14
11	1.16	0.66	0.16	0.42	0.18	0.49	1.51	0.19
12	1.08	1.28	0.16	0.42	0.17	0.47	1.57	0.19
13	1.03	0.73	0.13	0.36	0.14	0.40	1.44	0.16
14	1.33	1.41	0.21	0.38	0.18	0.51	2.31	0.22
15	1.18	1.35	0.22	0.34	0.16	0.52	2.74	0.20
Mean (*n* = 15)	1.19	1.02	0.19	0.39	0.18	0.52	1.90	0.21
SD	0.19	0.39	0.06	0.05	0.05	0.12	0.55	0.06
RSD	0.16	0.38	0.31	0.14	0.25	0.23	0.29	0.27

The active ingredients in 15 different batches of IRR were analyzed by LC-MS, and their relative molecular masses were compared with the standard substance. Eight compounds were identified: L-arginine (21.2%), guanine (18.2%), l—phenylalanine (3.4%), epigoitrin (7.0%), deoxidization vasicinone (3.2%), 3—indole acetonitrile (9.3%), indigo (33.9%), and indigo jade red (3.8%).

**Table 2 animals-14-02776-t002:** Analysis of Major Amino Acid Content in the IRR.

Ingredient	Content (%)	Ingredient	Content (%)
ASP	0.59	VAL	0.39
GLU	0.80	MET	0.05
SER	0.28	PHE	0.31
HIS	0.22	ILE	0.28
GLY	0.34	LEU	0.46
THR	0.30	LYS	0.37
ARG	0.94	PRO	0.73
Total	6.63		

**Table 3 animals-14-02776-t003:** Ingredient compositions and nutrient levels of the basal diet (as-fed basis).

Ingredient	Content (%)				
IRR	CON	0.5	1	2	4
Corn	56.08	55.63	55.08	53.88	51.68
Fermented soybean meal	30.1	30	30	30	29.8
Soybean oil	3	3.05	3.1	3.3	3.7
Glucose	2	2	2	2	2
Sucrose	3	3	3	3	3
Limestone	1	1	1	1	1
CaHPO_4_	1.5	1.5	1.5	1.5	1.5
NaCl	0.3	0.3	0.3	0.3	0.3
Citric acid	0.9	0.9	0.9	0.9	0.9
Choline chloride	0.1	0.1	0.1	0.1	0.1
L-Lysine HCl	0.6	0.6	0.6	0.6	0.6
DL-Methionine	0.3	0.3	0.3	0.3	0.3
L-Threonine	0.12	0.12	0.12	0.12	0.12
Premix	1	1	1	1	1
Total	100	100	100	100	100
Nutrient Levels	Content				
	CON	0.5	1	2	4
Digestible energy, kcal/kg	3509	3509	3506	3506	3506
Crude protein, %	19.82	19.81	19.81	19.83	19.82
Calcium, %	0.8	0.8	0.8	0.8	0.8
Total phosphorus, %	0.6	0.6	0.6	0.6	0.6
Available phosphorus, %	0.38	0.38	0.38	0.38	0.38
SID lysine, %	1.45	1.45	1.45	1.45	1.45
SID methionine, %	0.56	0.56	0.56	0.56	0.56
SID methionine + cystine, %	0.76	0.76	0.76	0.76	0.76
SID threonine, %	0.78	0.78	0.78	0.78	0.78
SID tryptophan, %	0.19	0.19	0.19	0.19	0.19

The premix provided the following nutrients per kilogram of diet: Cu (126.00 mg), Fe (102.00 mg), Zn (106.50 mg), Mn (17.70 mg), I (0.18 mg), Se (0.14 mg), VA (8000 U), VB1 (1.8 mg), VB2 (4.4 mg), VB6 (4.4 mg), VB12 (0.025 mg), VC (150.00 mg), VD (1000.00 U), 25-OH-D (0.025 mg), VE (120.00 mg), pantothenic acid (12.40 mg), niacinamide (25.00 mg), folic acid (0.88 mg), and biotin (132.00 mg). The standard digestible amino acids (SIDs) are expressed as calculated values, while the rest of the nutrients are measured values.

**Table 4 animals-14-02776-t004:** Primers used for gene expression analysis by real-time PCR.

Gene	Primer Sequences (5′-3′)	Size, bp
IL-1β	F: CCTGGACCTTGGTTCTCT	123
	R: GGATTCTTCATCGGCTTCT	
IL-10	F: TCGGCCCAGTGAAGAGTTTC	127
	R: GGAGTTCACGTGCTCCTTGA	
IL-6	F: AAATGTCGAGGCCGTGCAGATTAG	86
	R: GGGTGGTGGCTTTGTCTGGATTC	
TNF-α	F: ACAGGCCAGCTCCCTCTTAT	102
	R: CCTCGCCCTCCTGAATAAAT	
ZO-1	F: TTGATAGTGGCGTTGACA	126
	R: CCTCATCTTCATCATCTTCTAC	
Claudin-1	F: GCATCATTTCCTCCCTGTT	97
	R: TCTTGGCTTTGGGTGGTT	
Occludin	F: CAGTGGTAACTTGGAGGCGTCTTC	103
	R: CGTCGTGTAGTCTGTCTCGTAATGG	
β-actin	F: CTGCGGCATCCACGAAACT	147
	R: AGGGCCGTGATCTCCTTCTG	

IL-β = interleukin-β, IL-10 = interleukin-10, IL-6 = interleukin-6, TNF-α = tumor necrosis factor-α.

## Data Availability

The datasets presented in this study can be found in online repositories. The name(s) of the repository/repositories and accession number(s) can be found below: NCBI, PRJNA1011898.

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
