# Peer review of "Dietary Isatidis Root Residue Improves Diarrhea and Intestinal Function in Weaned Piglets"

_animals, 2024, doi:10.3390/ani14192776_

Round 1

Reviewer 1 Report

Comments and Suggestions for Authors

Dietary Isatidis root residues improve diarrhea and intestinal function in weaned piglets. ANIMALS

Very interesting research topic. On the one hand, searching for natural solutions to health problems of pigs kept in large-scale farm conditions and the need to limit antibiotics usage due to growing antibiotics resistance, and on the other hand, management of by-products. The topic fits well with current issues of circular economy and "zero waste" production, as well as the farm animals health improvement.

-Line 26: „growth performance” is repeated twice

-L29: the abbreviation „IRR” has no explanation here.

-Keywords: I would add „intestinal morphology” and „tight junction”.

-Introduction: too much about the Chinese medicine in general, there is no information about the actual experimental factor: Isatidis root – it’s activity/properties, bioactive substances, effects in animals/pigs, etc. It would be useful to have information about root residues, how it is obtained/produced.

-L74-80: whether chapter “2.1. Materials” describes the process how the root residues is obtained as a by-product during the production of drugs, or the preparation of residues for introduction into the feed mixture for pigs?

-chapter 2.3. Animals and dietary treatments: information is necessary on how the pigs were kept (conditions) or whether the pigs were fed ad libitum. In Table 2, the composition and nutritional value of the mixtures for all experimental groups must be added. After adding 0.5 to 4% root residues, were the mixtures balanced as isoprotein and isoenergetic?

-Discussion: do not use the term “Chinese herbal medicine”, as it is too general and does not explain anything (CHM administration - what does it mean?). Please provide the names of specific herbs and its quantities in the feed mixture used in the cited literature.

-Discussion and Conclusions: the text lacks data on the composition and nutritional value of the feed mixtures for all groups, so it is not known whether the experimental mixtures contained a higher level of fiber or not (and maybe other nutrients). This may have been the direct cause of the observed results, including slightly lower feed intake and body weight gains (although P > 0.05).

The manuscript needs major revision.

Author Response

Line 26: „growth performance” is repeated twice

Response: Thank you for your reminder. We have revised it.

L29: the abbreviation „IRR” has no explanation here.

Response: Thanks. We have revised it.

Keywords: I would add „intestinal morphology” and „tight junction”.

Response: Thank you. We believe your suggestion is excellent, and we have included "intestinal morphology" and "tight junction" in the keywords.

Introduction: too much about the Chinese medicine in general, there is no information about the actual experimental factor: Isatidis root – it’s activity/properties, bioactive substances, effects in animals/pigs, etc. It would be useful to have information about root residues, how it is obtained/produced.

Response: We greatly appreciate your suggestion, which we believe is highly beneficial. In accordance with your feedback, we have removed the content pertaining to herbal medicine from the article. Furthermore, we have revised and expanded the section on the application of isatis root in animals. (Line 52-80)

-L74-80: whether chapter “2.1. Materials” describes the process how the root residues is obtained as a by-product during the production of drugs, or the preparation of residues for introduction into the feed mixture for pigs?

Response: We deeply appreciate your suggestion. In accordance with your feedback, we have added details regarding the source and preparation method of Isatidis root residues in Section 2.1, Materials. (Line 85 -90)

chapter 2.3. Animals and dietary treatments: information is necessary on how the pigs were kept (conditions) or whether the pigs were fed ad libitum.

Response: We appreciate your suggestion. In response, we have provided additional information and clarification in the article regarding the free feeding of piglets during the experiment. (Line 132-143)

In Table 2, the composition and nutritional value of the mixtures for all experimental groups must be added.

Response: We would like to express our gratitude for your feedback. In accordance with your suggestion, we have revised and updated Table 2 in the original article.

After adding 0.5 to 4% root residues, were the mixtures balanced as isoprotein and isoenergetic?

Response: We appreciate your feedback. In this study, isatis root residue was utilized as the raw material. Accordingly, the supplementary data pertaining to the feed prepared after calculations for each experimental group are presented in Table 2.

Discussion: do not use the term “Chinese herbal medicine”, as it is too general and does not explain anything (CHM administration - what does it mean?).

Response: We appreciate your feedback. In response, we have revised the mention of Chinese herbal medicine in the original text and provided a detailed listing of the specific Chinese herbal medicines utilized. Furthermore, we clarify that the original intention behind CHM administration refers to the application of Chinese herbs. (Table 2)

Discussion and Conclusions: the text lacks data on the composition and nutritional value of the feed mixtures for all groups, so it is not known whether the experimental mixtures contained a higher level of fiber or not (and maybe other nutrients). This may have been the direct cause of the observed results, including slightly lower feed intake and body weight gains (although P > 0.05).

Response: Thank you very much for summarizing our article. Based on your feedback, we have supplemented the feed nutrient content for each group. (Table 2)

Reviewer 2 Report

Comments and Suggestions for Authors

The aim of this work was to investigate the potential of Isatidis root residue as a non-traditional feed resource. More detail can be provided in some article areas because the study covered a wide spectrum of expertise and a large amount of data was collected.

Lines 60–65. The introduction should include a detailed explanation of the selection of this plant, the active substances that are still present, and any instances in which the plant or its leftovers have been used in animal husbandry.

Lines 99–106 A detailed description should be given of the method, sample preparation for chromatography analysis, chromatographic conditions, and chromatograph type.

Lines 104-112 How was the feed supplemented with the additive?

Lines 217-226. Unclear. You wrote „...compared to the control piglets, although this increase was not statistically significant (P > 0.05) (Figure 1B)," but in Figure 1B it seems significant. Perhaps you might clarify what the "a, b, ab" letters in your Figure 1B mean.

Lines 228–231. Inaccurate description of the data. For instance, you  wrote, “Furthermore, dietary supplementation with 0.5% IRR led to a reduction in serum TBA, LDL, and GLB compared to the control group (P < 0.05)." However, Figure 2 shows that GLB was reduced in all treatment groups.

Lines 303-306. Figure 6 is too small; it's not possible to analyze data.

Lines 390-396. Conclusions are too general; please indicate which supplement concentration was most appropriate.

Author Response

Lines 60–65. The introduction should include a detailed explanation of the selection of this plant, the active substances that are still present, and any instances in which the plant or its leftovers have been used in animal husbandry.

Response: We sincerely appreciate your valuable advice. In response to your feedback, we have revised the content by reducing the focus on Chinese herbal medicine and instead emphasizing the active ingredients present in isatis root, along with exploring the application of these ingredients in animal husbandry. (Lines 52-80)

Lines 99–106 A detailed description should be given of the method, sample preparation for chromatography analysis, chromatographic conditions, and chromatograph type.

Response: We appreciate your suggestion. In accordance with your feedback, we have enhanced the original article by providing additional information and clarification on the preparation, conditions, and types of samples utilized for chromatographic analysis. (Lines 93 -125)

Lines 104-112 How was the feed supplemented with the additive?

Response: We sincerely appreciate your feedback. Based on the basic nutritional value of isatidis root, various proportions of isatidis root residue were incorporated into the feed formulation according to the calculated ratios. (Table 2)

Lines 217-226. Unclear. You wrote „...compared to the control piglets, although this increase was not statistically significant (P > 0.05) (Figure 1B)," but in Figure 1B it seems significant. Perhaps you might clarify what the "a, b, ab" letters in your Figure 1B mean.

Response: Thank you for your reminder. We have revised it. (Lines 249-252)

Lines 228–231. Inaccurate description of the data. For instance, you  wrote, “Furthermore, dietary supplementation with 0.5% IRR led to a reduction in serum TBA, LDL, and GLB compared to the control group (P < 0.05)." However, Figure 2 shows that GLB was reduced in all treatment groups.

Response: We appreciate your reminder and have made the necessary revisions accordingly (Lines 261-272).

Lines 303-306. Figure 6 is too small; it's not possible to analyze data.

Response: We appreciate your suggestion and have accordingly enlarged the graph.

Lines 390-396. Conclusions are too general; please indicate which supplement concentration was most appropriate.

Response: We sincerely appreciate your invaluable feedback. Following your recommendation, we have incorporated the suggested optimal dose range of 0.5-1% into the conclusion section of the article.

Round 2

Reviewer 1 Report

Comments and Suggestions for Authors

The Authors have taken advantage of all my comments and have significantly improved their manuscript. The article is suitable for publication.

Author Response

The Authors have taken advantage of all my comments and have significantly improved their manuscript. The article is suitable for publication.

Response: Thank you sincerely for your kind words and positive feedback regarding our manuscript.  We are pleased to learn that our diligent incorporation of your invaluable comments has resulted in a substantial enhancement of the article's quality.

Reviewer 2 Report

Comments and Suggestions for Authors

I appreciate the responses you have given. I suggest placing all of the chromatographic criteria in a single paragraph as opposed to listing them in sections (lines 98-130).

Author Response

Lines 99–106 A detailed description should be given of the method, sample preparation for chromatography analysis, chromatographic conditions, and chromatograph type.

Response: Thank you for your suggestion. I have consolidated all the mass spectrometry criteria into a single paragraph. (lines 93-111).